# New Cycloadditon Reaction of 2-Chloroprop-2-enethioamides with Dialkyl Acetylenedicarboxylates: Synthesis of Dialkyl 2-[4,5-bis(alkoxycarbonyl)-2-(aryl{alkyl}imino)-3(2*H*)-thienylidene]-1,3-dithiole-4,5-dicarboxylates

**DOI:** 10.3390/molecules27206887

**Published:** 2022-10-14

**Authors:** Vladimir A. Ogurtsov, Oleg A. Rakitin

**Affiliations:** N. D. Zelinsky Institute of Organic Chemistry, Russian Academy of Sciences, 119991 Moscow, Russia

**Keywords:** sulfur heterocycles, 3*H*-1,2-dithiole-3-thiones, dialkyl 2-(2-amino-1-chloro-2-thioxoethylidene)-1,3-dithiole-4,5-dicarboxylates, 1,3-dipolar cycloaddition, DMAD, amines, thioamides

## Abstract

The 1,3-dipolar cycloaddition of 1,2-dithiole-3-thiones with alkynes to form 1,3-dithioles is one of the most studied reactions in this class of polysulfur-containing heterocycles. Nucleophilic substitution of chlorine atoms in dimethyl 2-(1,2-dichloro-2-thioxoethylidene)-1,3-dithiole-4,5-dicarboxylate, which was obtained by addition one molecules of DMAD to 4,5-dichloro-3*H*-1,2-dithiole-3-thione, led to a series of 2-chloro-2-(1,3-dithiol-2-ylidene)ethanethioamides. Cycloaddition reaction of 2-chloro-2-(1,3-dithiol-2-ylidene)ethanethioamides with activated alkynes led to the unexpected formation of 2-(thiophen-3(2*H*)-ylidene)-1,3-dithioles via new intermediate, 1-(1,3-dithiol-2-ylidene)-*N*-phenylethan-1-yliumimidothioate. Structure of dimethyl 2-(4,5-bis(methoxycarbonyl)-2-(phenylimino)thiophen-3(2*H*)-ylidene)-1,3-dithiole-4,5-dicarboxylate was finally proven by single crystal X-ray diffraction study. Optimized reaction conditions and a mechanistic rationale for the 1,3-dipolar cycloaddition of novel intermediate are presented.

## 1. Introduction

1,2-Dithiole-3-thiones **1** have been investigated since 1884, when their first representative, 4,5-dimethyl-3*H*-1,2-dithiole-3-thione, was isolated [1]. According to Scifinder and Reaxys search, more than 2000 compounds of this class together their complexes have been obtained to date. There are two reasons for such a rapid development of these heterocycles. First of all, 1,2-dithiole-3-thiones have many important biological activities (see, reviews [2,3]); this ring is present in many commercial drugs, such as Oltipraz [4], anethole dithiolethione (ADT) [5], S-Danshensu [6], and NOSH-1 [7]. The second reason for interest in 1,2-dithiole-3-thiones is the rich chemistry of these heterocycles [8], which can lead to compounds that enhance the nonlinear optical properties to create organic electronic conductors [9], photoconductive materials [10,11] and semiconducting polymers [12]. The most typical reaction of 1,2-dithiole-3-thiones, which has been studied for decades, is 1,3-dipolar cycloaddition to alkynes, as a rule containing electron-withdrawing groups. It is known that one or two alkyne molecules are sequentially added to 1,2-dithiole-3-thiones by the [3 + 2] cycloaddition reactions, first giving 1,3-dithioles **2** and then spiro-1,3-dithiolothiopyrans **3** (Figure 1).

The presence of electron-accepting groups, such as chlorine [13], fluorine and trifluoromethyl groups [14], in the 1,2-dithiole-3-thione suppressed the second cycloaddition of alkynes, while spiro-1,3-dithiolothiopyrans **3** did not form. The addition product of one alkyne to 4,5-dichloro-3*H*-1,2-dithiole-3-thione **4** reacted with the second alkyne with molecular rearrangement and loss of chlorine to give 7*H*-thieno [2,3-*c*]thiopyran-7-thiones **6** and 4*H*-thieno [3,2-*c*]thiopyran-4-thiones **7** (Figure 2).

Thioacyl chloride **5a** easily reacted with some amines, phenol and thiophenol to form the corresponding thioamides, thiono and dithio esters **8** in high yields (Figure 3) [15].

Herein, we describe the synthesis of thioacyl amides **9** and its cycloaddition reaction with activated alkynes to unexpectedly form of thiophen-3(2*H*)-ylidenes **10** via new intermediate **14**.

## 2. Results and Discussion

The nucleophilic substitution of dichlorothioxoethylidenedithiole **5a** has been poorly studied. Known examples are reactions with benzylamine, *m*-toluidine, morpholine, phenol and thiophenol in acetone at room temperature with the formation of the corresponding thioamides and thiono and dithio esters **8**, usually in high yields [15]. We have studied in detail the reactivity of nitrogen nucleophiles with compound **5a**. Treatment of this compound with a twofold excess of aniline in MeCN gave thioamide **9a** in good yield (Table 1, Entry 1). The use of DMF instead of acetonitrile as a solvent shortened the reaction time and slightly increased the yield of final product **9a** (Entry 2). The most logical and economical in terms of amine consumption seemed to be the use of one mole of a tertiary amine, i.e., trimethylamine, to bind the hydrogen chloride molecule, which is released upon reaction with the amine. It turned out that the replacement of one mole of aniline to Et_3_N or Cs_2_CO_3_ led to almost the same yield of product **9a** (Entries 3 and 4). Similar patterns were noted when these methods were extended to other aromatic and aliphatic amines; moderate to high yields of thioamides **9** were achieved (Entries 5, 8 and 20). For *p*-nitroaniline, similar conditions also turned out to be optimal, but in this case it was necessary to heat the reaction mixture to 80 °C (Entry 15). We have shown that the often more accessible and stable hydrochlorides or hydrobromides of aromatic and aliphatic amines can be successfully introduced into this reaction (Entries 7, 18 and 22).

The reaction of thioamides **9** with dimethyl acetylenedicarboxylate (DMAD) was thoroughly studied (Table 2). Treatment of anilino derivative **9a** with DMAD at elevated temperature afforded, in various solvents (o-xylene, MeCN or DMF), novel compound **10a** as a yellow solid C_21_H_17_NO_8_S_3_ (Entries 1, 3 and 5). There was no reaction in any solvent at room temperature (Entries 2,4). The reaction in xylene requires many hours of heating (55 h) until the starting compound **9a** disappears from the reaction mixture (TLC CH_2_Cl_2_/petroleum ether–10/1). The use of acetonitrile as a solvent requires not only an elevated temperature, but also the presence of a base, triethylamine (Entry 3). Heating of the starting compounds in DMF turned out to be the most convenient; the yield of the final compound was the highest 78% (Entry 5).

According to the mass spectrum, elemental analysis, and ^1^H and ^13^C NMR data (Appendix A), compound **10a** is formally a product of DMAD addition and HCl elimination. Several isomeric structures can be proposed for this product; analysis of the obtained data did not allow us to confidentially choose the structure for this compound. The structure **10a** was finally confirmed by single crystal X-ray diffraction study as dimethyl 2-(4,5-bis(methoxycarbonyl)-2-(phenylimino)thiophen-3(2*H*)-ylidene)-1,3-dithiole-4,5-dicarboxylate (Figure 1).

The reaction of thioamides **9** with DMAD was extended to give thiophen-3(2*H*)-ylidenes **10** in moderate yields (Table 2, entries 6–10). Curiously, thioamide **9g**, containing dimethylthiomide group and not containing NH group, did not react with DMF even in harsh conditions (100 °C, 10 h) gradually decomposing into products, from which it was not possible to isolate individual compounds. When a mixture of thioamide **9g** and DMAD was irradiated in a ratio of 1:2 in a solution of chlorobenzene with an SVD-120 UV lamp (power 120 W) for 70 min at a temperature of 25–37 °C, no new products were detected (TLC); the starting compound **9g** was isolated in almost quantitative yield.

Replacing DMAD with diethyl acetylenedicarboxylate (DEAD) in the reaction with thioamide **9a** in DMF at 100 °C for 2 h yielded a product containing two dimethylcarboxy and two diethylcarboxyethyl groups **11a** in good yield (Figure 4). Its isomer **11b** was obtained by treating diethyl 2-(1-chloro-2-(phenylamino)-2-thioxoethylidene)-1,3-dithiole-4,5-dicarboxylate **12** with DMAD under the same conditions with almost the same yield.

The synthesis of thiophen-3(2*H*)-ylidenes **10** can be carried out in one pot from readily available 4,5-dichloro-3*H*-1,2-dithiole-3-thione **4**, which is a precursor for the preparation of thioxoethylidenedithiole **5a** [13]. DMAD (1 equiv.) was successively added to a solution of dithiolethione **4** and the reaction mixture was stirred for 8 h at room temperature, then amine (2 equiv.) with stirring for 2 h at room temperature, and finally DMAD with heating at 100 °C for 3 h (Figure 5). Compounds **10** were isolated in moderate yields. In the case of 4-nitroaniline (**d**), the time of the second step was increased to 10 h, and in case **e**, methylamine hydrochloride was used and the addition of Et_3_N (1 eqv) was necessary.

Thus, 1,3-dipolar cycloaddition of thioamides **9** with DMAD led to the formation of dihydrothiophene ring in **10** with HCl extrusion. One of the possible pathways for this unusual transformation is suggested in Figure 6. In the reaction of thioamide **9a** with an alkyne molecule, the presence of non-reactive thioamide group can suppress the Diels-Alder cycloaddition to adduct **13** similarly to indole derivatives [16] in favor of extrusion of HCl under the action of a base (Me_2_NH from DMF or Et_3_N) to form active betaine **14**. This extrusion became impossible for disubstituted thioamides (i.e., **9g**) and the slow decomposition of this compound was observed. Intermediate **14** readily reacted as 1,3-dipole with DMAD to form dihydrothiophene **10**.

2-(Thiophen-3(2*H*)-ylidene)-1,3-dithiole ring system is rare, according to Reaxys and SciFinder search, there are several examples of these analogs. They named heteroquinoid merocyanine dyes and investigated in vacuum-processed organic solar cells [17].

## 3. Experimental Section

### 3.1. Materials and Reagents

The chemicals were purchased from the commercial sources (Sigma-Aldrich, St. Louis, MO, USA) and used as received. Dimethyl 2-(1,2-dichloro-2-thioxoethylidene)-1,3-dithiole-4,5-dicarboxylate **3** [15], diethyl 2-(1,2-dichloro-2-thioxoethylidene)-1,3-dithiole-4,5-dicarboxylate **7 [13]** and 4,5-dichloro-3*H*-1,2-dithiole-3-thione **8** [18], were prepared according to the previously described protocols. All synthetic operations were performed under a dry argon atmosphere. The solvents were purified by distillation over the appropriate drying agents.

### 3.2. Analytical Instruments

The melting points were determined on a Kofler hot-stage apparatus and were uncorrected. ^1^H and ^13^C NMR spectra were taken with a Bruker AM-300 machine (Bruker Ltd., Moscow, Russia) with TMS as the standard. *J* values are given in Hz. MS spectra (EI, 70 eV) were obtained with a Finnigan MAT INCOS 50 instrument (Thermo Finnigan LLC, San Jose, CA, USA). High-resolution MS spectra were measured on a Bruker micrOTOF II instrument using electrospray ionization (ESI). IR spectra were measured with a Bruker “Alpha-T” instrument (Bruker, Billerica, MA, USA) in KBr pellets.

### 3.3. X-ray Analysis

X-ray diffraction data were collected at 100K on a four-circle Rigaku Synergy S diffractometer equipped with a HyPix600HE area-detector (kappa geometry, shutterless ω-scan technique), using graphite monochromatized Cu K_α_-radiation. The intensity data were integrated and corrected for absorption and decay by the CrysAlisPro program [19]. The structure was solved by direct methods using SHELXT and refined on *F^2^* using SHELXL-2018 [20] in the OLEX2 program [21]. All non-hydrogen atoms were refined with individual anisotropic displacement parameters. All hydrogen atoms were placed in ideal calculated positions and refined as riding atoms with relative isotropic displacement parameters. The Cambridge Crystallographic Data Centre contains the supplementary crystallographic data for this paper No. CCDC 2205782 (**10a**). These data can be obtained free of charge via http://www.ccdc.cam.ac.uk/conts/retrieving.html (or from the CCDC, 12 Union Road, Cambridge CB2 1EZ, UK; Fax: +44-1223-336033; E-mail: deposit@ccdc.cam.ac.uk). Crystal data and structure refinement for this compound are given in Table 3.

### 3.4. General Procedure for the Synthesis of Thioamides 9a–g

Amine **a**,**b**,**c**,**f** (2 mmol) or amine **d**,**e**,**g** (1 mmol) together with Et_3_N (0.277 mL, 2 mmol) were added to a solution of thiochloride **5a** (345 mg, 1 mmol) in DMF (10 mL), and the reaction mixture was stirred at the temperature and for the time indicated in Table 1, then poured into water (60 mL) and stirred for 30 min, extracted with ethyl acetate (2 × 30 mL). Combined organic extracts were washed with dilute HCl (10%, 30 mL), brine (2 × 30 mL), dried with MgSO_4_. The solvent was removed under reduced pressure, and the residue was purified by column chromatography on silica gel (Silica gel Merck 60, eluent CH_2_Cl_2_/petroleum ether–10/1).

*Dimethyl 2-(1-chloro-2-(phenylamino)-2-thioxoethylidene)-1,3-dithiole-4,5-dicarboxylate* (**9a**)

Yellow crystals (340 mg, 85%), mp. 149–150 °C. R_f_ = 0.43 (CH_2_Cl_2_/petroleum ether–10/1). ^1^H NMR (300 MHz, CDCl_3_, δ, ppm): 3.92 (s, 3H), 3.94 (s, 3H), 7.32 (d, *J* = 7.3, 1H), 7.44 (t, *J* = 7.7, 2H), 7.60 (d, *J* = 8.1, 2H), 9.29 (s, 1H). ^13^C NMR (75 MHz, CDCl_3_, δ, ppm): 53.6, 106.7, 124.2, 127.0, 129.1, 129.2, 136.7, 138.4, 154.2, 159.5, 160.4, 180.8. MS (EI, 70 Ev), *m*/*z* (I, %): 403 (M + 2, 21), 402 (M + 1, 11), 401 (M^+^, 54), 370 (15), 368 (32), 365 (12), 311 (25), 309 (57), 259 (33), 226 (19), 194 (20), 159 (13), 135 (30), 100 (20), 77 (64), 59 (100), 15 (62). HRMS-ESI (*m*/*z*): calcd for (C_15_H_12_ClNO_4_S_3_) [M + H]^+^ 401.9690, found 401.9697. IR, ν, cm^−1^: 3435, 3348, 2951, 1730, 1718, 1589, 1519, 1462, 1440, 1430, 1352, 1251, 1195, 1090, 1012, 961, 822, 774, 747, 571, 537, 498, 473.

*Dimethyl 2-(1-chloro-2-thioxo-2-(p-tolylamino)ethylidene)-1,3-dithiole-4,5-dicarboxylate* (**9b**)

Yellow crystals (361 mg, 87%), mp. 127–128 °C. R_f_ = 0.42 (CH_2_Cl_2_/petroleum ether–10/1). ^1^H NMR (300 MHz, CDCl_3_, δ, ppm): 2.39 (s, 3H), 3.92 (s, 3H), 3.93 (s, 3H), 7.24 (d, *J* = 8.1, 2H), 7.45 (d, *J* = 8.1, 2H), 9.24 (s, 1H). ^13^C NMR (75 MHz, CDCl_3_, δ, ppm):21.2, 53.6, 106.6, 124.3, 129.1, 129.7, 135.8, 136.7, 137.1, 153.9, 159.5, 160.4, 180.9. MS (EI, 70 Ev), *m*/*z* (I, %): 417 (M + 2, 32), 416 (M + 1, 17), 415 (M^+^, 68), 384 (24), 382 (55), 379 (18), 311 (48), 309 (100), 291 (16), 273 (31), 240 (12), 208 (19), 173 (12), 135 (22), 91 (24), 59 (156), 15 (32). HRMS-ESI (*m*/*z*): calcd for (C_16_H_14_ClNO_4_S_3_) [M + H]^+^ 415.9846, found 415.9833. IR, ν, cm^−1^: 3436, 3345, 2950, 2361, 1742, 1715, 1581, 1518, 1490, 1431, 1340, 1283, 1229, 1094, 1018, 960, 911, 823, 737, 698, 631, 551, 501, 490.

*Dimethyl 2-(1-chloro-2-((4-methoxyphenyl)amino)-2-thioxoethylidene)-1,3-dithiole-4,5-dicarboxylate* (**9c**)

Yellow crystals (310 mg, 72%), mp. 159–160 °C. R_f_ = 0.35 (CH_2_Cl_2_/petroleum ether–10/1). ^1^H NMR (300 MHz, CDCl_3_, δ, ppm): 3.84 (s, 3H), 3.91 (s, 3H), 3.93(s, 3H), 6.95 (d, *J* = 8.8, 2H), 7.44 (d, *J* = 9.5, 2H), 9.18 (s, 1H). ^13^C NMR (75 MHz, CDCl_3_, δ, ppm): 53.6, 55.5, 106.6, 114.2, 126.2, 129.0, 131.3, 136.7, 138.4, 153.8, 158.4, 160.4, 181.1. MS (EI, 70 Ev), *m*/*z* (I, %): 433 (M + 2, 20), 432 (M + 1, 16), 431 (M^+^, 44), 400 (25), 398 (41), 311 (34), 310 (8), 309 (78), 274 (22), 226 (8), 225 (4), 224 (16), 189 (15), 174 (12), 140 (100), 135 (41), 122 (38), 95 (26), 59 (78), 15 (52). HRMS-ESI (*m*/*z*): calcd for (C_16_H_14_ClNO_5_S_3_) [M + H]^+^ 431.9795, found 431.9807. IR, ν, cm^−1^: 3435, 3333, 2955, 2837, 1741, 1726, 1592, 1534, 1509, 1473, 1432, 1365, 1288, 1235, 1180, 1026, 828, 814, 652, 534, 518.

*Dimethyl 2-(1-chloro-2-((4-nitrophenyl)amino)-2-thioxoethylidene)-1,3-dithiole-4,5-dicarboxylate* (**9d**)

Yellow crystals (272 mg, 61%), mp. 198–199 °C. R_f_ = 0.33 (CH_2_Cl_2_/petroleum ether–10/1). ^1^H NMR (300 MHz, CDCl_3_, δ, ppm): 3.93 (s, 3H), 3.95 (s, 3H), 7.93 (d, *J* = 9.5, 2H), 7.44, 8.29 (d, *J* = 9.5, 2H), 9.42 (s, 1H). ^13^C NMR (75 MHz, DMSO-d_6_, δ, ppm): 53.8, 106.7, 118.7, 124.0, 125.9, 129.9, 134.5, 140.1, 145.4, 159.0, 159.9, 181.8. MS (EI, 70 Ev), *m*/*z* (I, %): 448 (M + 2, 8), 447 (M + 1, 2), 446 (M^+^, 30), 415 (3), 414 (1), 413 (8), 311 (4), 309 (11), 241 (3), 240 (1), 239 (8), 226 (2), 193 (4), 146 (7), 135 (32), 100 (17), 84 (24), 59 (100), 15 (64). HRMS-ESI (*m*/*z*): calcd for (C_15_H_11_ClN_2_O_6_S_3_) [M + H]^+^ 446.9541, found 446.9543. IR, ν, cm^−1^: 3436, 3324, 3116, 2954, 1727, 1710, 1593, 1575, 1548, 1503, 1467, 1434, 1335, 1289, 1204, 1189, 1034, 900, 849, 823, 750, 705, 659, 476.

*Dimethyl 2-(1-chloro-2-(methylamino)-2-thioxoethylidene)-1,3-dithiole-4,5-dicarboxylate* (**9e**)

Yellow crystals (261 mg, 77%), mp. 128–129 °C. R_f_ = 0.37 (CH_2_Cl_2_/petroleum ether–10/1). ^1^H NMR (300 MHz, CDCl_3_, δ, ppm): 3.21 (d, *J* = 4.4, 3H), 3.91 (s, 3H), 3.93 (s, 3H), 7.89 (br s, 1H). ^13^C NMR (75 MHz, CDCl_3_, δ, ppm): 32.8, 53.5, 106.5, 128.7, 136.1, 152.0, 159.6, 160.4, 183.2. MS (EI, 70 Ev), *m*/*z* (I, %): 341 (M + 2, 38), 340 (M + 1, 14), 339 (M^+^, 78), 280 (14), 272 (24), 244 (15), 199 (40), 198 (8), 197 (100), 164 (6), 132 (54), 100 (32), 84 (34), 74 (52), 59 (92), 15 (76). HRMS-ESI (*m*/*z*): calcd for (C_10_H_10_ClNO_4_S_3_) [M + H]^+^ 339.9533, found 339.9539, calcd for (C_10_H_10_ClNO_4_S_3_) [M + Na]^+^ 361.9353, found 361.9354, calcd for (C_10_H_10_ClNO_4_S_3_) [M + K]^+^ 377.9092, found 377.9092. IR, ν, cm^−1^: 3435, 3365, 2955, 2340, 1728, 1706, 1584, 1522, 1491, 1430, 1293, 1234, 1090, 1024, 821, 765, 561, 475.

*Dimethyl 2-(2-(tert-butylamino)-1-chloro-2-thioxoethylidene)-1,3-dithiole-4,5-dicarboxylate* (**9f**)

Yellow crystals (210 mg, 55%), mp. 61–62 °C. R_f_ = 0.49 (CH_2_Cl_2_/petroleum ether–10/1). ^1^H NMR (300 MHz, CDCl_3_, δ, ppm): 1.58 (s, 9H), 3.89 (s, 3H), 3.91 (s, 3H), 7.73 (br s, 1H)). ^13^C NMR (75 MHz, CDCl_3_, δ, ppm): 27.8, 53.4, 55.6, 106.6, 128.4, 136.6, 151.2, 159.7, 160.6, 181.2. MS (EI, 70 Ev), *m*/*z* (I, %): 483 (M + 2, 38), 482 (M + 1, 18), 481 (M^+^, 71), 367 (22), 366 (14), 365 (52), 325 (22), 311 (24), 310 (7), 309 (55), 293 (82), 239 (18),183(60), 135 (14), 100 (14), 59 (64), 57 (100), 41(74), 29 (46), 15 (36). HRMS-ESI (*m*/*z*): calcd for (C_13_H_16_ClNO_4_S_3_) [M + H]^+^ 382.003, found 381.9999, calcd for (C_13_H_16_ClNO_4_S_3_) [M + Na]^+^ 403.9822, found 403.9802, calcd for (C_13_H_16_ClNO_4_S_3_) [M + K]^+^ 419.9562, found 419.9556. IR, ν, cm^−1^: 3435, 3360, 2954, 1731, 1590, 1519, 1476, 1431, 1363, 1253, 1207, 1170, 1094, 1017, 911, 812, 763, 662, 553, 482.

*Dimethyl 2-(1-chloro-2-(dimethylamino)-2-thioxoethylidene)-1,3-dithiole-4,5-dicarboxylate* (**9g**)

Yellow crystals (219 mg, 62%), mp. 55–56 °C. R_f_ = 0.31 (CH_2_Cl_2_/petroleum ether–10/1).^1^H NMR (300 MHz, CDCl_3_, δ, ppm): 3.35 (s, 6H), 3.84 (s, 3H), 3.86 (s, 3H). ^13^C NMR (75 MHz, CDCl_3_, δ, ppm): 43.3, 53.4, 106.5, 129.2, 133.5, 140.2, 159.5, 159.8, 190.2. MS (EI, 70 Ev), *m*/*z* (I, %): 355 (M + 2, 22), 354 (M + 1, 6), 353 (M^+^, 40), 337 (4), 318 (8), 286 (12), 275 (22), 258 (10), 229 (8), 214 (8), 176 (6), 147 (20), 135 (20), 112 (30), 100 (36), 88 (50), 59 (94), 42 (56), 29 (28), 15 (100). HRMS-ESI (*m*/*z*): calcd for (C_11_H_12_ClNO_4_S_3_) [M + H]^+^ 353.9690, found 353.9701, calcd for (C_11_H_12_ClNO_4_S_3_) [M + Na]^+^ 375.9509, found 375.9511, calcd for (C_11_H_12_ClNO_4_S_3_) [M + K]^+^ 391.9249, found 391.9257. IR, ν, cm^−1^: 3427, 2960, 1722, 1588, 1536, 1509, 1453, 1435, 1382, 1254, 1172, 1138, 1094, 1019, 961, 822, 760, 747, 694, 466.

*Diethyl 2-(1-chloro-2-(phenylamino)-2-thioxoethylidene)-1,3-dithiole-4,5-dicarboxylate* (**12**)

Phenylamine (0.273 mL, 3 mmol) was added to a solution of diethyl 2-(2-anilino-1-chloro-2-thioxoethylidene)-1,3-dithiole-4,5-dicarboxylate **7** (560 mg, 1.5 mmol) in DMF (15 mL) and the reaction mixture was stirred at 25 °C, then poured into water (80 mL) and stirred for 30 min, extracted with ethyl acetate (2 × 40 mL). Combined organic extracts were washed with dilute HCl (10%, 40 mL), brine (2 × 40 mL), dried with MgSO_4_. The solvent was removed under reduced pressure and the residue was purified by column chromatography on silica gel (Silica gel Merck 60, eluent CH_2_Cl_2_/petroleum ether–10/1).

Yellow crystals (502 mg, 78%), mp. 103–104 °C. R_f_ = 0.48 (CH_2_Cl_2_/petroleum ether–10/1). ^1^H NMR (300 MHz, CDCl_3_, δ, ppm): 1.40 (m, 6H), 4.39 (m, 4H), 7.31 (m, 1H), 7.44 (t, *J* = 7.9, 2H), 7.60 (d, *J* = 8.1, 2H), 9.29 (s, 1H). ^13^C NMR (75 MHz, CDCl_3_, δ, ppm): 14.0, 63.0, 106.5, 124.2, 127.0, 127.8, 129.0, 136.9, 138.3, 154.5, 159.0, 160.1, 180.1. MS (EI, 70 Ev), *m*/*z* (I, %): 431 (M + 2, 5), 430 (M + 1, 1), 429 (M^+^, 12), 396 (4), 337 (8), 259 (10), 237 (61), 191 (12), 175 (18), 135 (6), 97 (6), 84 (10), 77 (14), 55 (23), 29 (100), 16 (39). HRMS-ESI (*m*/*z*): calcd for (C_17_H_16_ClNO_4_S_3_) [M + H]^+^ 430.0003, found 429.9993, calcd for (C_17_H_16_ClNO_4_S_3_) [M + Na]^+^ 451.9822, found 451.9810. IR, ν, cm^−1^: 3453, 3304, 2923, 1731, 1719, 1590, 1517, 1466, 1439, 1317, 1258, 1239, 1194, 1090, 1018, 820, 744, 692, 624, 553, 459.

### 3.5. General Procedure for the Synthesis of Thiophen-3(2H)-ylidenes from Thioamides

A solution of thioamide **9a**–**f**,**12** (0.5 mmol) and DMAD or DEAD (1 mmol) in DMF (7 mL) was stirred at 100 °C for the time indicated in Table 2 or for 2 h, then cooled, poured into water (50 mL) and stirred for 30 min, extracted with ethyl acetate (2 × 30 mL). Combined organic extracts were washed with dilute HCl (10%, 30 mL), brine (2 × 30 mL), dried with MgSO_4_. The solvent was removed under reduced pressure, and the residue was purified by column chromatography on silica gel (Silica gel Merck 60, eluent CH_2_Cl_2_/petroleum ether–10/1).

*Dimethyl 2-(4,5-bis(methoxycarbonyl)-2-(phenylimino)thiophen-3(2H)-ylidene)-1,3-dithiole-4,5-dicarboxylate* (**10a**)

Dark red crystals (198 mg, 78%), mp. 165–166 °C. R_f_ = 0.20 (CH_2_Cl_2_/petroleum ether–10/1). ^1^H NMR (300 MHz, DMSO-d_6_, δ, ppm): 3.76 (s, 3H), 3.85 (s, 3H), 3.87 (s, 3H), 3.97 (s, 3H), 7.15 (d, *J* = 7.5, 2H), 7.23 (m, 1H), 7.46 (t, *J* = 7.8, 2H). ^13^C NMR (75 MHz, DMSO-d_6_, δ, ppm): 53.5, 54.2, 54.3, 54.4, 119.3, 120.8, 122.0, 125.9, 130.1, 132.0, 133.2, 136.2, 149.4, 152.5, 158.3, 159.2, 159.9, 161.2, 165.4. MS (EI, 70 Ev), *m*/*z* (I, %): 507 (M^+^, 100), 491 (5), 476 (6), 448 (40), 398 (9), 286 (5), 243 (5), 112 (8), 77 (11), 59 (18), 44 (10), 32 (8), 15 (16 HRMS-ESI (*m*/*z*): calcd for (C_21_H_17_NO_8_S_3_) [M + H]^+^ 508.0189, found 508.0186, calcd for (C_21_H_17_NO_8_S_3_) [M + Na]^+^ 530.0008, found 530.0004, calcd for (C_21_H_17_NO_8_S_3_) [M + K]^+^ 545.9748, found 545.9748. IR, ν, cm^−1^: 3423, 2949, 1728, 1720, 1568, 1561, 1461, 1460, 1240, 1205, 1073, 757, 481.

*Dimethyl 2-(4,5-bis(methoxycarbonyl)-2-(p-tolylimino)thiophen-3(2H)-ylidene)-1,3-dithiole-4,5-dicarboxylate* (**10b**)

Red crystals (198 mg, 70%), mp. 141–142 °C. R_f_ = 0.19 (CH_2_Cl_2_/petroleum ether–10/1). ^1^H NMR (300 MHz, CDCl_3_, δ, ppm): 2.39 (s, 3H), 3.84 (s, 3H), 3.94 (s, 6H), 4.08 (s, 3H), 7.11 (d, *J* = 8.1, 2H), 7.24 (d, *J* = 8.1, 2H), 7.60. ^13^C NMR (75 MHz, CDCl_3_, δ, ppm): 21.2, 52.9, 53.8, 53.8, 53.8, 120.8, 123.7, 129.6, 130.2, 133.3, 135.3, 146.7, 150.4, 158.1, 159.9, 160.6, 162.1, 166.1, 166.2. MS (EI, 70 Ev), *m*/*z* (I, %): 521 (M^+^, 100), 490 (6), 462 (53), 398 (7), 286 (4), 59 (7), 28 (4). HRMS-ESI (*m*/*z*): calcd for (C_22_H_19_NO_8_S_3_) [M + H]^+^ 522.0346, found 522.0333, calcd for (C_22_H_19_NO_8_S_3_) [M + Na]^+^ 544.0165, found 544.0169, calcd for (C_22_H_19_NO_8_S_3_) [M + K]^+^ 559.9904, found 559.9904. IR, ν, cm^−1^: 3434, 2951, 2926, 1731, 1578, 1547, 1460, 1430, 1345, 1247, 1069, 824, 813, 761, 489, 473.

*Dimethyl 2-(4,5-bis(methoxycarbonyl)-2-((4-methoxyphenyl)imino)thiophen-3(2H)-ylidene)-1,3-dithiole-4,5-dicarboxylate* (**10c**)

Dark red crystals (180 mg, 67%), mp. 147–148 °C. R_f_ = 0.13 (CH_2_Cl_2_/petroleum ether–10/1). ^1^H NMR (300 MHz, CDCl_3_, δ, ppm): 3.83 (s, 3H), 3.84 (s, 3H), 3.93 (s, 6H), 4.07 (s, 3H), 6.97 (d, *J* = 8.8, 2H), 7.18 (d, *J* = 8.8, 2H). ^13^C NMR (75 MHz, CDCl_3_, δ, ppm): 52.9, 53.8, 53.8, 55.6, 114.8, 120.7, 122.2, 123.6, 133.3, 142.4, 149.9, 157.2, 157.6, 159.9, 162.8, 166.2. MS (EI, 70 Ev), *m*/*z* (I, %): 537 (M^+^, 95), 522 (16), 506 (6), 478 (58), 398 (25), 331 (15), 317 (14), 286 (28), 268 (56), 243 (15), 143 (28), 112 (83), 92 (40), 77 (42), 59 (100), 44 (56), 15 (51). HRMS-ESI (*m*/*z*): calcd for (C_22_H_19_NO_9_S_3_) [M + H]^+^ 538.0295, found 538.0279, calcd for (C_22_H_19_NO_9_S_3_) [M + Na]^+^ 560.0114, found 560.0102, calcd for (C_22_H_19_NO_9_S_3_) [M + K]^+^ 575.9854, found 575.9854. IR, ν, cm^−1^: 3605, 3435, 2950, 1728, 1711, 1605, 1564, 1503, 1466, 1430, 1345, 1245, 1201, 1179, 1098, 1071, 825, 758, 691, 584, 480.

*Dimethyl 2-(4,5-bis(methoxycarbonyl)-2-((4-nitrophenyl)imino)thiophen-3(2H)-ylidene)-1,3-dithiole-4,5-dicarboxylate* (**10d**)

Dark red crystals (160 mg, 58%), mp. 179–180 °C. R_f_ = 0.15 (CH_2_Cl_2_/petroleum ether–10/1). ^1^H NMR (300 MHz, CDCl_3_, δ, ppm): 3.86 (s, 3H), 3.97 (s, 6H), 4.09 (s, 3H), 7.30 (m, 2H), 8.31 (d, *J* = 8.8, 2H). ^13^C NMR (75 MHz, CDCl_3_, δ, ppm): 53.1, 54.0, 54.0, 120.5, 121.5, 125.6, 130.7, 133.5, 135.5, 139.2, 144.5, 150.5, 154.8, 159.6, 160.1, 161.5, 162.7, 166.6. MS (EI, 70 Ev), *m*/*z* (I, %): 552 (M^+^, 64), 522 (6), 494 (31), 398 (3), 286 (7), 143 (8), 112 (29), 94 (8), 75 (15), 59 (100), 30 (24), 15 (23). HRMS-ESI (*m*/*z*): calcd for (C_22_H_16_N_2_O_10_S_3_) [M + H]^+^ 553.0040, found 553.0040. IR, ν, cm^−1^: 3439, 3066, 2955, 1762, 1713, 1585, 1553, 1510, 1452, 1431, 1328, 1243, 1176, 1104, 1066, 999, 924, 844, 758, 690, 483.

*Dimethyl 2-(4,5-bis(methoxycarbonyl)-2-(methylimino)thiophen-3(2H)-ylidene)-1,3-dithiole-4,5-dicarboxylate* (**10e**)

Orange crystals (162 mg, 73%), mp. 107–108 °C. R_f_ = 0.12 (CH_2_Cl_2_/petroleum ether–10/1). ^1^H NMR (300 MHz, CDCl_3_, δ, ppm): 3.47 (s, 3H), 3.87 (s, 3H), 3.92 (s, 3H), 3.94 (s, 3H), 4.06 (s, 3H). ^13^C NMR (75 MHz, CDCl_3_, δ, ppm): 42.4, 51.5, 51.8, 52.1, 52.8, 124.7, 127.4, 130.1, 138.5, 140.2, 156.4, 160.4, 164.8, 165.2, 165.6, 165.9. MS (EI, 70 Ev), *m*/*z* (I, %): 445 (M^+^, 12), 386 (8), 328 (2), 286 (3), 243 (2), 228 (3), 185 (4), 112 (11), 97 (16), 83 (22), 69(32), 59 (71), 43 (100), 29 (31), 15 (40). HRMS-ESI (*m*/*z*): calcd for (C_16_H_15_NO_8_S_3_) [M + H]^+^ 446.0033, found 446.0032. IR, ν, cm^−1^: 3079, 2924, 2853, 1718, 1641, 1605, 1542, 1465, 1435, 1394, 1364, 1346, 1248, 1190, 1082, 1024, 991, 909, 722, 474.

*Dimethyl 2-(2-(tert-butylimino)-4,5-bis(methoxycarbonyl)thiophen-3(2H)-ylidene)-1,3-dithiole-4,5-dicarboxylate* (**10f**)

Orange oil (157 mg, 63%). R_f_ = 0.15 (CH_2_Cl_2_/petroleum ether–10/1). ^1^H NMR (300 MHz, CDCl_3_, δ, ppm): 1.41 (d, *J* = 2.9, 9H), 3.87 (s, 3H), 3.88 (s, 3H), 3.91 (s, 3H), 3.93 (s, 3H). ^13^C NMR (75 MHz, CDCl_3_, δ, ppm): 29.45, 54.1, 54.2, 54.4, 54.5, 119.8, 121.8, 134.4, 137.3, 137.6, 152.1, 155.6, 161.5, 161.9, 164.9, 165.9. MS (EI, 70 Ev), *m*/*z* (I, %): 487 (M^+^, 3), 472 (5), 429 (2), 399 (5), 365 (8), 339 (7), 309 (13), 293 (10), 235 (7), 176 (2), 119 (8), 93 (8), 84 (11), 57 (100), 41 (64), 29 (46), 15 (56). HRMS-ESI (*m*/*z*): calcd for (C_19_H_21_NO_8_S_3_) [M + H]^+^ 488.0502, found 488.0501. IR, ν, cm^−1^: 3635, 3428, 2956, 2921, 1728, 1634, 1593, 1521, 1434, 1364, 1343, 1251, 1099, 1068, 1024, 913, 817, 767, 688, 480.

*Dimethyl 2-(4,5-bis(ethoxycarbonyl)-2-(phenylimino)thiophen-3(2H)-ylidene)-1,3-dithiole-4,5-dicarboxylate* (**11a**)

Dark red crystals (206 mg, 77%), mp. 83–85 °C. R_f_ = 0.21 (CH_2_Cl_2_/petroleum ether–10/1). ^1^H NMR (300 MHz, CDCl_3_, δ, ppm): 1.32 (m, 3H), 1.49 (m, 3H), 3.92 (s, 3H), 3.94 (s, 3H), 4.30 (q, *J* = 7.3, 2H), 4.56 (m, 2H), 7.21 (m, 3H), 7.44 (t, *J* = 7.7, 2H). ^13^C NMR (75 MHz, CDCl_3_, δ, ppm): 13.9, 14.2, 53.1, 53.3, 62.0, 62.1, 120.1, 120.8, 120.9, 125.1, 125.4, 129.6, 130.5, 135.8, 149.5, 151.8, 159.2, 160.2, 161.4, 161.6, 165.7. MS (EI, 70 Ev), *m*/*z* (I, %): 535 (M^+^, 4), 477 (5), 286 (2), 228 (2), 175 (3), 125 (2), 77 (7), 59 (6), 29 (100), 15 (4). HRMS-ESI (*m*/*z*): calcd for (C_23_H_21_NO_8_S_3_) [M + H]^+^ 536.0502, found 536.0499. IR, ν, cm^−1^: 3436, 3078, 2924, 1727, 1711, 1580, 1552, 1461, 1367, 1248, 1197, 1072, 1019, 909, 760, 694, 476.

*Diethyl 2-(4,5-bis(methoxycarbonyl)-2-(phenylimino)thiophen-3(2H)-ylidene)-1,3-dithiole-4,5-dicarboxylate* (**11b**)

Dark red crystals (201 mg, 75%), mp. 184–185 °C. R_f_ = 0.17 (CH_2_Cl_2_/petroleum ether–10/1). ^1^H NMR (300 MHz, CDCl_3_, δ, ppm): 1.40 (m, 6H), 3.84 (s, 3H), 4.08 (s, 3H), 4.40 (m, 4H), 7.21 (m, 3H), 7.44 (t, *J* = 7.3, 2H). ^13^C NMR (75 MHz, CDCl_3_, δ, ppm): 14.0, 52.9, 53.7, 63.4, 119.4, 120.9, 125.1, 125.4, 129.6, 129.9, 135.5, 137.6, 149.4, 152.1, 157.9, 158.8, 160.2, 161.9, 166.1. MS (EI, 70 Ev), *m*/*z* (I, %): 535 (M^+^, 16), 463 (11), 434 (7), 352 (6), 301 (3), 270 (7), 214 (10), 172 (8), 143 (11), 125 (12), 111 (18), 97 (42), 83 (44), 71 (49), 59 (48), 55 (73), 43 (100), 29 (69), 15 (28). HRMS-ESI (*m*/*z*): calcd for (C_23_H_21_NO_8_S_3_) [M + H]^+^ 536.0502, found 536.0490. IR, ν, cm^−1^: 3535, 3078, 2924, 1728, 1641, 1581, 1465, 1434, 1366, 1347, 1252, 1208, 1082, 1026, 909, 762, 694, 476.

### 3.6. General Procedure for the Synthesis of Thiophen-3(2H)-ylidenes 10a–f from 4,5-dichloro-3H-1,2-dithiole-3-thione 4

A solution of 4,5-dichloro-3*H*-1,2-dithiole-3-thione **4** (203 mg, 1 mmol) and DMAD (0.123 mL, 1 mmol) in DMF (10 mL) was stirred at 25 °C for 8 h. Amine (2mmol) is added to the reaction mixture and stirred at the same temperature for another 2 h. Then, a new portion of DMAD (0.245 mL, 2 mmol) is added to the reaction mass and stirred at a temperature was stirred at 100 °C for 3h, then cooled, poured into water (80 mL) and stirred for 30 min, extracted with ethyl acetate (2 × 40 mL). Combined organic extracts were washed with dilute HCl (10%, 30 mL), brine (2 × 40 mL), dried with MgSO_4_. The solvent was removed under reduced pressure, and the residue was purified by column chromatography on silica gel (Silica gel Merck 60, eluent CH_2_Cl_2_/petroleum ether–10/1). Yields are given in Figure 5.

## 4. Conclusions

2-Chloro-2-(1,3-dithiol-2-ylidene)ethanethioamides, synthesised from readily available dimethyl 2-(1,2-dichloro-2-thioxoethylidene)-1,3-dithiole-4,5-dicarboxylate, were found to undergo 1,3-cycloaddition reaction with activated alkynes to form 2-imino-3(2*H*)-thiophenylidenes. Structure of 2-(thiophen-3(2*H*)-ylidene)-1,3-dithioles was finally proved by single crystal X-ray diffraction study. The unusual formation of a five-membered dihydrothiophene ring from 2-chloro-2-(1,3-dithiol-2-ylidene)ethanethioamides can be explained by the formation of new betainic intermediate, 1-(1,3-dithiol-2-ylidene)-*N*-phenylethan-1-yliumimidothioate. Thiophene-3(2*H*)-ylidene)-1,3-dithiole can be considered as a valuable scaffold of heteroquinoid merocyanine dyes for use in vacuum processed organic solar cells.

## Data Availability

The data presented in this study are available in Appendix A.

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
