# Peer review of "New Cycloadditon Reaction of 2-Chloroprop-2-enethioamides with Dialkyl Acetylenedicarboxylates: Synthesis of Dialkyl 2-[4,5-bis(alkoxycarbonyl)-2-(aryl{alkyl}imino)-3(2H)-thienylidene]-1,3-dithiole-4,5-dicarboxylates"

_molecules, 2022, doi:10.3390/molecules27206887_

Round 1

Reviewer 1 Report

Comments:

In this manuscript, the Prof. Oleg A. Rakitin and coworkers reported several kinds of dialkyl 2-[4,5-bis(alkoxycarbonyl)-2-(aryl{alkyl}imino)-3(2H)-thienylidene]-1,3-dithiole-4,5-dicarboxylates were synthesis by cycloaddition reaction of dialkyl 2-(2-amino-1-chloro-2-thioxoethylidene)-1,3-dithiole-4,5-dicarboxylates, and the products structure finally proven by single crystal X-ray diffraction study. The manuscript is novel, results are impressive, and the manuscript can be considered for publication; however, there are some minor comments:

1.    Please added the reaction conditions in Scheme 1, 2, 3and 4.

2.    Table 1 is not very clear to read, and please divide table 1 into two tables. First is the Optimization of the Reaction Conditions. Second is the nucleophilic substitution scope of dimethyl 2-(1,2-dichloro-2-thioxoethylidene)-1,3-dithiole-4,5-dicarboxylate 3 by nitrogen nucleophiles.

3.    Page 6, line 125. “In the case of 4-nitroaniline (e), the time of the second step was increased to 10 h, and in case f, methylamine hydrochloride was used 126 and the addition of Et3N (1 eqv) was necessary.”, 4-nitroaniline (e) should be 4-nitroaniline (d), and in case f should be in case e.

4.    Page 8, line 181. “Amine 9a, b, c, f (2 mmol) or amine 9 d, e, g (1 mmol) together with Et3N (0.277 mL, 2 mmol”, the aniline 9a, b, c, d, e, f should be the products, not the start material. Please check the remark these anilines number.

5.    Please check the compound 10b, it’s seems lost one CH3?

6.    Please draw the chemical structure on the spectrum.

7.    Some compounds H NMR and C NMR are not good, such as: 9d, 10f, 11a, 11b, please provide the high-quality 1H and 13C NMR spectra.

Author Response

Response to Reviewer 1.

The authors are grateful to the reviewer for a highly professional review.

Reviewer 1:

Please added the reaction conditions in Scheme 1, 2, 3 and 4.

Authors:

The reaction conditions in Scheme 1 are too different for different radicals and cannot be combined, so the reaction conditions were added to Schemes 2-4.

Reviewer 1:

Table 1 is not very clear to read, and please divide table 1 into two tables. First is the Optimization of the Reaction Conditions. Second is the nucleophilic substitution scope of dimethyl 2-(1,2-dichloro-2-thioxoethylidene)-1,3-dithiole-4,5-dicarboxylate 3 by nitrogen nucleophiles.

Authors:

Unfortunately, the authors had to optimaze the reaction conditions for the nucleophilic substitution reaction not only for aniline, but also for all other amines and their hydrochlorides. Therefore, it is not possible to divide Table 1 into two parts.

Reviewer 1:

Page 6, line 125. “In the case of 4-nitroaniline (e), the time of the second step was increased to 10 h, and in case f, methylamine hydrochloride was used 126 and the addition of Et3N (1 eqv) was necessary.”, 4-nitroaniline (e) should be 4-nitroaniline (d), and in case f should be in case e.

Authors:

Corrected as requested by the Reviewer.

Reviewer 1:

Page 8, line 181. “Amine 9a, b, c, f (2 mmol) or amine 9 d, e, g (1 mmol) together with Et3N (0.277 mL, 2 mmol”, the aniline 9a, b, c, d, e, f should be the products, not the start material. Please check the remark these anilines number.

Authors:

Corrected as requested by the Reviewer.

Reviewer 1:

Please check the compound 10b, it’s seems lost one CH3?

Authors:

The integral signal intensity of the signal 3.94 ppm in 1H NMR spectrum turned out to be equal to 6 protons. Corrected as requested by the Reviewer.

Reviewer 1:

Please draw the chemical structure on the spectrum.

Authors:

Added.

Reviewer 1:

Some compounds H NMR and C NMR are not good, such as: 9d, 10f, 11a, 11b, please provide the high-quality 1H and 13C NMR spectra.

Authors:

Corrected as requested by Reviewer.

Reviewer 2 Report

The manuscript "New cycloadditon reaction of 2-chloroprop-2-enethioamides with dialkyl acetylenedicarboxylates: synthesis of dialkyl 2-[4,5-bis(alkoxycarbonyl)-2-(aryl{alkyl}imino)-3(2H)-thienylidene]-1,3-dithiole-4,5-dicarboxylates" by Ogurtsov and Rakitin reports on the synthesis of heterocycles 10a-f (presumably) via a [2+3] dipolar cycloaddition. The mechanism is plausible, though unproven.

This reviewer found reading the introduction difficult. The authors should make explicitly clear what has been done in the past (either by them or by others), and what is novel, for example by explicitly naming the authors of past works. Here is what I think:
Already known:
-) 1,2-Dithiole-3-thiones 1 form compounds 2 and subsequently compounds 3 with electron-poor alkynes if the substituents of 1 are electron-donating.
-) If the substituents of 1 are electron-withdrawing (compounds 4), then 3 does not form. Instead, either isomeric structures 6,7 can be synthesized, or the intermediate 5 (equivalent to 2) can be used as a precursor for thioamides 8 via nucleophilic substitution.
Novel:
1) Several analogs of substances 8 have been produced: 9a-g
2) If those substances contain a thioamidic N-H bond (9a-f), they can undergo the titular cycloaddition with an electron-poor alkyne (DMAD or DEAD here; do you think others work as well? Please comment) to yield compounds 10a-f.
The structures of compounds 10 were elucidated by X-ray crystallography of compound 10a.
3) The authors have developed a one-pot procedure starting with 1,2-Dithiole-3-thione 4 and producing compounds 10 directly (hence compounds 9 occur only as intermediates).

In the opinion of this reviewer, this manuscript could be published in Molecules and in particular in this Special Issue. However, the readability of the manuscript should be improved to increase its potential impact.

Comments:

1) In order to make the abstract, main text, conclusions (and perhaps the title) more readable I suggest replacing the full systematic names either just by the compound numbers or by the name of the substance class, for instance
"dialkyl 2-[4,5-bis(alkoxycarbonyl)-2-(aryl{alkyl}imino)-3(2H)-thienylidene]-1,3-dithiole-4,5-dicarboxylates" -> "thiophen-3(2H)-ylidenes"
"new intermediate, 1-(1,3-dithiol-2-ylidene)-N-phenylethan-1-yliumimidothioate 14" -> "new intermediate 14"

2) Please add the mechanisms of the reactions shown in Schemes 1 and 2, if known.

3) Line 110, Figure 1 caption: I think the compound is mislabeled; you mean compound 10a, not 5a?

4) Scheme 6: The authors propose a plausible dipolar cycloaddition mechanism. However, how is Cl- eliminated? Perhaps the addition of DMAD occurs first, stabilizing a negative charge at S---C(-)---S in the left-hand heterocycle, then eliminate the Cl-?

5) The authors have provided the 1H and 13C NMR spectra in the SI. I recommend adding the mass spectra as well.

6) Why do the two carboxymethyl groups of compounds 9 and the four carboxymethyl groups of compounds 10 show slightly different shifts?

7) Since the structure elucidation of compounds 10 relied on X-ray crystallography, please provide the CIF report in the supplementary materials.

Author Response

Response to Reviewer 2.

The authors are grateful to the reviewer for a kind and highly professional review.

Reviewer 2:

In order to make the abstract, main text, conclusions (and perhaps the title) more readable I suggest replacing the full systematic names either just by the compound numbers or by the name of the substance class, for instance "dialkyl 2-[4,5-bis(alkoxycarbonyl)-2-(aryl{alkyl}imino)-3(2H)-thienylidene]-1,3-dithiole-4,5-dicarboxylates" -> "thiophen-3(2H)-ylidenes" "new intermediate, 1-(1,3-dithiol-2-ylidene)-N-phenylethan-1-yliumimidothioate 14" -> "new intermediate 14"

Authors:

The necessary changes have been made to the abstract, main text, and conclusions as requested by the reviewer.

Reviewer 2:

Please add the mechanisms of the reactions shown in Schemes 1 and 2, if known.

Authors:

An explanation of the reaction mechanism given in Scheme 1 has been added to the text of the paper. The mechanism of the reactions described in Scheme 2 is too complex and voluminous, so we considered it necessary not to include it in the paper.

Reviewer 2:

Line 110, Figure 1 caption: I think the compound is mislabeled; you mean compound 10a, not 5a?

Authors:

Corrected as requested by the Reviewer.

Reviewer 2:

Scheme 6: The authors propose a plausible dipolar cycloaddition mechanism. However, how is Cl- eliminated? Perhaps the addition of DMAD occurs first, stabilizing a negative charge at S---C(-)---S in the left-hand heterocycle, then eliminate the Cl-?

Authors:

The reaction mechanism proposed by the reviewer can also be considered. However, we believe that the first step in the reaction is the elimination of hydrogen chloride, either thermally (in xylene) or catalyzed by bases. One of the proofs of this is the fact that the reaction with the dimethylamine derivative 9g, in which HCl cleavage is impossible, does not occur, but the starting compound 9g is isolated.

Reviewer 2:

The authors have provided the 1H and 13C NMR spectra in the SI. I recommend adding the mass spectra as well.

Authors:

Added

Reviewer 2:

Why do the two carboxymethyl groups of compounds 9 and the four carboxymethyl groups of compounds 10 show slightly different shifts?

Authors:

As follows from the X-ray diffraction data for compound 10a, compounds 10 exist in one regioform with respect to the double bond between two heterocycles, which is the reason for the nonequivalence of the two carboxymethyl groups in the 1,3-dithiole ring.

Reviewer 2:

Since the structure elucidation of compounds 10 relied on X-ray crystallography, please provide the CIF report in the supplementary materials.

Authors:

Crystal data and structure refinement for compound 10a (No. CCDC 2205782) are added to the Supplementary Materials.

Reviewer 3 Report

It  is   good  scientific  study  but  it  needs Minor  corrections  to be  acceptable  in this  journal :

1- The  abstract  is very brief ,, it  needs  more addition  of results 

2- There  is  no   ratio  of  solvent  that used  in  TLC in  experiments 

3- There  is  no  compared between  results in current  study   in  results  in prevousily  studies  to  improve  current  results 

4- replace  old references  No (1   and  18 )  by other more updated 

5- There  is  no   Clear explanation   for  results  in  Table (2)  . 

6- There  are a lot of Self  citation  in  references No. (8 , 13 , 15)

7- I accepted  paper  after  Minor  Corrections 

Author Response

Response to Reviewer 3.

The authors are grateful to the reviewer for a kind and highly professional review.

Reviewer 3:

The abstract is very brief, it needs more addition of results

Authors:

Abstract expanded.

Reviewer 3:

There is no ratio of solvent that used in TLC in experiments

Authors:

Added.

Reviewer 3:

There is no compared between results in current study in results in prevousily studies to improve current results

Authors:

The introduction contains references to cycloaddition reactions of 1,2-dithiole-3-thiones containing halogen atoms with alkynes containing electron-withdrawing groups. These works use either heating (ref. 13) or UV irradiation (ref. 14). Our article describes the behavior of reaction mixtures when heated. We also carried out reactions under UV irradiation (the relevant material has been added to the text of the article, line 110). However, we failed to obtain significant results in this direction.

Reviewer 3:

replace old references No (1 and 18) by other more updated

Authors:

Reference 1 is a historical reference and we cannot change it. Reference 18 has been replaced for more updated one.

Reviewer 3:

There is no Clear explanation for results in Table (2).

Authors:

Minor changes have been made to the notes to Table 2 on pages 4 and 5.

Reviewer 3:

There are a lot of Self citation in references No. (8 , 13 , 15)

Authors:

Reference 8 has been replaced, references 13 and 15 cannot be replaced, as they are devoted to a special part of chemistry.

Reviewer 4 Report

This manuscript described the synthesis of thioacyl amides 9 and its cycloaddition reaction with alkynes. It is well written, the authors discussed the mechanism, and all compounds gave clear NMR, I am glad the authors also provide x-ray data. I have a few questions listed here.

1.     Line 63, The use of DMF instead of acetonitrile as a solvent shortened the reaction time and slightly increased the yield of 64 final product 9a (Entry 2).

How is the solubility of the reaction, did all reactants dissolve in CH3CN?

2.     Line 67, 68, It turned out that the replacement of one mole of aniline to Et3N led to almost the same yield of product 9a (Entry 3).

This probably is because of the solvent effect, DMF.

That is why I am curious about the reactant solubility, if reactants were not well dissolved in CH3CN, then entry 4, only using 1 equivalent of aniline, not adding Et3N, would have the same result, in terms of yield and reaction time.

Especially, by reading Table 1, compare entries 9,12,13.

3.     The authors did not try the base, Cs2CO3, which is a very good base in terms of solubility, you only need a weak base to trap the HCl, for reaction in Table 1.

4.     Reactant 9 reacts with DMAD if uses o-dichlorobenzene as the solvent, and is heated to 150°C, which would give a very good cycloaddition reaction.

5.     From Line 112 to 116, compare the reaction of 9a with DEAD VS the reaction of 12 with DMAD, which should include the reaction condition, solvent, reaction time, heat, or rt.

6.     Line 181 to 184, the procedure was described wrong, compounds 9 were products, in Table 1.

7.     Table 1, title was wrong, dimethyl 2-(1,2-dichloro-2-thioxoethylidene)-1,3-dithiole-4,5- 76 dicarboxylate was compound 5, not 3.

8.     The proton NMR should be 0ppm-13ppm.

Author Response

Response to Reviewer 4.

The authors are grateful to the reviewer for a kind and highly professional review.

Reviewer 4:

  1. Line 63, The use of DMF instead of acetonitrile as a solvent shortened the reaction time and slightly increased the yield of 64 final product 9a (Entry 2). How is the solubility of the reaction, did all reactants dissolve in CH3CN?

Authors:

All reagents are perfectly soluble in organic solvents. Obviously, the shortening of the time in DMF is due to the fact that the dipolarity of this solvent facilitates the 1,3-dipolar cycloaddition reaction.

Reviewer 4:

  1. Line 67, 68, It turned out that the replacement of one mole of aniline to Et3N led to almost the same yield of product 9a (Entry 3). This probably is because of the solvent effect, DMF. That is why I am curious about the reactant solubility, if reactants were not well dissolved in CH3CN, then entry 4, only using 1 equivalent of aniline, not adding Et3N, would have the same result, in terms of yield and reaction time. Especially, by reading Table 1, compare entries 9,12,13.

Authors:

All reagents are perfectly soluble in organic solvents. Obviously, the shortening of the time in DMF is due to the fact that the dipolarity of this solvent facilitates the 1,3-dipolar cycloaddition reaction.

Reviewer 4:

  1. The authors did not try the base, Cs2CO3, which is a very good base in terms of solubility, you only need a weak base to trap the HCl, for reaction in Table 1.

Authors:

Unfortunately, the addition of cesium carbonate to the reaction, as well as triethylamine, did not lead to an increase in the yield of the final product (see Entry 4 in the Table 1).

Reviewer 4:

  1. Reactant 9 reacts with DMAD if uses o-dichlorobenzene as the solvent, and is heated to 150°C, which would give a very good cycloaddition reaction.

Authors:

The results of the reaction in o-dichlorobenzene are added to Table 2.

Reviewer 4:

  1. From Line 112 to 116, compare the reaction of 9a with DEAD VS the reaction of 12 with DMAD, which should include the reaction condition, solvent, reaction time, heat, or rt.

Authors:

The necessary information has been added to the text of the article and to Scheme 4.

Reviewer 4:

  1. Line 181 to 184, the procedure was described wrong, compounds 9 were products, in Table 1.

Authors:

Corrected as requested by the Reviewer.

Reviewer 4:

  1. Table 1, title was wrong, dimethyl 2-(1,2-dichloro-2-thioxoethylidene)-1,3-dithiole-4,5- 76 dicarboxylate was compound 5, not 3.

Authors:

Corrected as requested by the Reviewer.

Reviewer 4:

  1. The proton NMR should be 0ppm-13ppm.

Authors:

Corrected as requested by the Reviewer.

Reviewer 5 Report

Oleg A. Rakitin and coworker reported the new cycloadditon reaction of 2-chloroprop-2-enethioamides with dialkyl acetylenedicarboxylates. They reported a new type of cycloaddition reaction with various new organic compounds and also proposed mechanism of this cycloaddition reaction. All the new compounds are well characterized. This work is well written and presented, therefore deserve for publication in Molecules in current form. 

Author Response

The authors are grateful to the reviewer for a kind and highly professional review. There is no issues in this reference.